# Octopus: A Novel Approach for Health Data Masking and Retrieving Using Physical Unclonable Functions and Machine Learning

**DOI:** 10.3390/s23084082

**Published:** 2023-04-18

**Authors:** Sagar Satra, Pintu Kumar Sadhu, Venkata P. Yanambaka, Ahmed Abdelgawad

**Affiliations:** 1College of Science and Engineering, Central Michigan University, Mount Pleasant, MI 48858, USA; 2Department of Mathematics and Computer Science, Texas Woman’s University, Denton, TX 76204, USA

**Keywords:** internet of medical things, physical unclonable functions, machine learning, security and privacy

## Abstract

Health equipment are used to keep track of significant health indicators, automate health interventions, and analyze health indicators. People have begun using mobile applications to track health characteristics and medical demands because devices are now linked to high-speed internet and mobile phones. Such a combination of smart devices, the internet, and mobile applications expands the usage of remote health monitoring through the Internet of Medical Things (IoMT). The accessibility and unpredictable aspects of IoMT create massive security and confidentiality threats in IoMT systems. In this paper, Octopus and Physically Unclonable Functions (PUFs) are used to provide privacy to the healthcare device by masking the data, and machine learning (ML) techniques are used to retrieve the health data back and reduce security breaches on networks. This technique has exhibited 99.45% accuracy, which proves that this technique could be used to secure health data with masking.

## 1. Introduction

A person’s health is critical for leading a peaceful and prosperous future. The World Health Organization (WHO) defines health as a condition of bodily and psychological well being free of sickness or disability [1]. The healthcare system maintains and enhances one’s health through isometric exercises, sickness diagnoses, and cures. Majority of traditional healthcare employs manual management and monitoring of patient demographic information, prior cases, diagnoses, prescription, invoicing, and pharmaceutical inventory upkeep, which results in human mistakes and negatively impacts patients. IoMT technology, mainly based on the Internet of Things (IoT), eliminates human errors. This assists physicians in diagnosing diseases more quickly and precisely by integrating all essential parameter-monitoring equipment through a connection with a systematic approach [2]. The IoMT combines the IoT with medical equipment. All clinical gadgets will be linked to and analyzed by healthcare experts through the internet in the future of IoMTs. This will enable quicker and less expensive medical care as it develops. Figure 1 displays a case of an IoMT in which the individual vital signs are gathered by sensor devices and transmitted to the IoMT applications accessible online. The knowledge is subsequently passed to the medical professionals and personnel, who respond and then communicate with the required patients [3].

The IoMT is also a collection of clinical technologies and applications that connect computer networks to the healthcare profession. In the recent decade, IoMT has received a lot of attention. IoMT is open to attack by innumerable appearing cyber-attacks, identity fraud, keylogging, phishing, and harmful bots, as well as information processing and security issues and data sharing across networks [4]. In 2015, through an analysis by HP Fortify, the ten leading smartwatches were all discovered to have security flaws. This is due to inadequate identity or permission, absence of transmitting data encryption, unsecured interfaces, vulnerable operating system, and privacy problems [5]. Validation, for instance, is confirming a user’s information. All approved and verified individuals or equipment must be able to acquire IoMT devices or wireless medical devices (WMDs). IoMT tools can be used for destruction in addition to abusing confidential information [6]. Inadequate identification safety might enable an attacker to get into the network and obtain access to a user’s personal health information. Medical devices with Wi-Fi networks enable machine-to-machine communication because it is at the heart of IoMT. Numerous methods are used to minimize the total cost of managing or preventing various serious illnesses. Healthcare Information solutions have progressed, allowing consumers to keep track of patients using networks.

The importance of individual and device identification to a network is that it guarantees that data are accurately ascribed. Moreover, the data in networks are only available to authorized personnel. Devices that continually monitor healthcare attributes, devices that automatically handle therapies, and devices that track real-time data while a patient self-manages a treatment are examples of devices with such data [7]. Another problem with WMDs is that they frequently depend on a private method for device and transmission link security. The IoMT smart devices are also a collection of sensors and electronic circuits. This helps to obtain the biomedical signals from the patient, network connectivity to send biomedical data via a network, and a base-band processor to analyze biomedical signals. Moreover, a short- or long-term storage unit and a display platform with artificial intelligence systems are used for making decisions based on the doctor’s availability [8]. The security objectives for IoMT devices can be divided into three categories: confidentiality, integrity, and availability. A full security analysis of these objectives is presented below:

Confidentiality: The safeguarding of sensitive data from unauthorized use is referred to as confidentiality. In the context of IoMT devices, this means preventing unauthorized individuals from seeing or stealing health information and other confidential material. Encryption, access control, and secure communication protocols are some of the security precautions that may be employed to ensure secrecy. Encryption may be used to safeguard data while it is in transit and at rest, ensuring that only authorized users have access to it. Password protection can be used to limit authorized users’ access to confidential data. Secure communication protocols, such as TLS, can be used to safeguard data transfer between IoMT devices and other systems, preventing data from being intercepted by illegal access.

Integrity: The protection of data from illegal alteration or manipulation is referred to as integrity. This involves ensuring that patient data and other sensitive information are not changed or updated without sufficient consent in the context of IoMT devices. Access control, intrusion detection and prevention, and secure boot are some of the security precautions that may be employed to ensure integrity. Access control can be used to limit authorized users’ exposure of confidential data, thus lowering the danger of unauthorized alteration. To identify and prevent unwanted data alteration, intrusion detection and prevention measures can be deployed. Secure boot can be used to guarantee that only trustworthy software are permitted to operate on IoMT devices, lowering the chance of malicious malware changing data.

Availability: The capacity of the system to be accessible and usable by authorized users is referred to as availability. This includes ensuring that the devices and data they manage are available when needed in the context of IoMT devices. Redundancy, backup and recovery, and physical security are just a few protection techniques that may be employed to ensure availability. Redundancy may be utilized to guarantee that, if an IoMT equipment fails, another device can take over and provide system stability. In the case of a system crash, backup and recovery procedures can be utilized to guarantee that data are not lost. Physical security measures can be used to secure IoMT equipment from physical harm and keep them operational.

In conclusion, meeting the security goals of confidentiality, integrity, and availability for IoMT devices necessitates a holistic strategy involving numerous security methods. Organizations may ensure the security of IoMT devices as well as the protection of patient data and other sensitive information by using these security features. Further detailed discussions on the proposed mechanism and goals are mentioned in the related research section.

### 1.1. IoMT and Smart E-Healthcare

The prevalence of internet-based computing in the healthcare industry has expanded the number of items that are connected. IoMT has applications in many fields, including smart cities, smart homes, remote health monitoring, smart healthcare, and power management. Here is a more in-depth description of IoMT and smart e-healthcare.

#### 1.1.1. IoMT and Enabling Wireless Technologies

Individuals, society, and doctors gain from IoMT in various ways. As with any technology, there are advantages and disadvantages. Wireless technology is used to transfer healthcare data with the use of the internet. The ample storage space provided by cloud services receives the unprocessed data acquired at these devices/sensors [9]. To obtain a further understanding of the data collected from the devices, it is further cleaned and then analyzed. This necessitates using new programs, devices, and tools that will improve the viewing, evaluation, transmission, and administration of the data [10].

Figure 2 depicts the interconnection of various wireless technologies, including NFC (Near Field Communications), LTE (Long Term Evolution), Bluetooth, RFID (Radio Frequency Identification), and 5G/6G (and beyond), with a variety of devices, including smartphones, multi-sensory bands, sensor systems, smart wearables, as well as other medical equipment [11]. Due to their enormous capacity and amazingly low latency advantages, 5G/6G and higher are currently widely used in IoMT.

#### 1.1.2. Smart E-Healthcare

Hospitals that use automatic and efficient modules (perhaps based on Artificial Intelligence/Machine Learning) on the Information and Communications Technology (ICT) infrastructure to enhance patient care processes and provide unique innovations are referred to as “smart hospitals”. Intelligent hospitals have a variety of uses, including telemedicine, telehealth, and remote robotic operation [12]. While telehealth focuses on providing non-clinical care remotely, telemedicine is used to deliver clinical treatment remotely. In remote robotic operations, medical robots carry out procedures under the direction of a surgeon who is located a great distance away.

### 1.2. Security and Privacy Requirements for IoMT Devices

IoMT equipment have stricter security and privacy requirements than ordinary IoT-based platforms. Figure 3 shows that the patient’s data should be handled privately [13]. If the data which are collected are breached, the person can be harassed, which can also lead the patient to be distressed and depressed. Much enhanced security is needed for IoMT medical systems, such as equipment positioning, which can help to secure the system’s security and privacy. Only authorized personnel should have access to healthcare information, which must be collected and stored in accordance with legal and ethical confidentiality. Appropriate steps must be taken to safeguard the integrity of health information linked with patient characteristics in order to avoid data intrusions. The necessity of such safeguards cannot be overstated since data obtained by cyber-criminals might be traded on black markets, putting patients at high risk of not just privacy breaches but also financial harm [14].

The goal of the data authenticity condition for IoMT healthcare systems is to verify that the information received at the desired target has still not been tampered with in any manner during the information transfer. Using the broadcasting feature of the wireless connection, intruders might obtain information and manipulate patient records, which could have severe consequences in hazardous situations [15]. To ensure that all information have still not been tampered with, the ability to identify any unlawful information alterations or modifications is essential. As a result, proper information safety must be developed to prevent unwanted attempts to alter sent information. Furthermore, the integrity of data maintained on medical devices must be guaranteed, which implies that information cannot be tampered with [16]. WMDs, particularly environmental sensors, can sometimes be seized, exposing sensitive data to hackers. Additionally, hackers can reconfigure the acquired equipment and re-deploy them to the network, monitoring conversations without really being detected [17]. As a result, medical equipment theft is a severe privacy problem that must be addressed and resolved in IoMT healthcare systems. Medical equipment in the networks must, at minimum, contain highly secured integrated circuits, which prohibit outsiders from reading codes placed on the equipment once they have been installed. Using PUFs to safeguard data collected in the Integrated Circuits (ICs) of medical equipment is one instance of an approach [18].

## 2. Related Research

Many innovative strategies for securing IoMT device data have been introduced in research articles. In recent studies, scientists claimed that cardiovascular heart disorders will kill around 23.6 million individuals by 2030. Lbrini et al. used ML to predict the cardiovascular risk present in the patient to alert them to doctors at an early stage. If danger emerges, the doctor will be notified, and they can offer diagnosis and guidance to the patients to avoid the health consequences [19]. This study will help patients to receive treatment as soon as possible, and it may save lives. However, this study did not cover the security aspect. The data can be manipulated and tampered with by an attacker to send out false information to the doctor, which can be risky. In another study, Zhao et al. analyzed hemiplegic gait based on wearable sensors. In the research, they used sensors which were worn on the patient’s waist and lower limbs to collect the data while they walked in a straight line. The data were processed and analyzed to claim that the methodology they used could be utilized to reconstruct the patient’s walking ability [20]. The security breach of the data can be avoided in both studies by masking the data. One study stated that cryptographic techniques must be implemented to secure communication systems among IoMT devices. There seem to be various designs and methods that can provide privacy against a variety of integrity flaws and risks, preventing illegal access to equipment. Symmetric and asymmetric methods can be used to limit access to the device and prevent an intruder from taking control of the system. These methods can help protect against spying assaults [21]. However, if the patient’s information is encrypted, it is possible that it could become a hurdle in a crisis. Whenever a healthcare professional needs to view device information, cryptographic solutions may limit access, putting the patient’s life in danger.

Alternatively, fair access codes can be given to certified medical professionals managing crises [22]. However, this contradicts the purpose of having cryptographic procedures in place since attackers may acquire access to the code and manipulate it with the IoMT device via a variety of means. These can protect from external attacks, but they are sensitive to strikes from short range. Moreover, by flashing ultraviolet light on the individual, an attacker can acquire access to the code [23,24]. The code is extracted from the gadget using electrocardiography data generated by the individual. The attack is known as “optical fault induction” or “optical glitching”, and it includes the use of a laser or other light source to alter the behavior of a device by leading it to misbehave. In some situations, an attacker may be able to extract sensitive information such as cryptographic keys or other data as a result of this. Nevertheless, numerous considerations make this sort of assault difficult to carry out on an IoMT device. To begin, the attacker would need physical access to the device, as well as specific tools and skills. Second, the gadget must be built to be susceptible to optical defect induction, which is not always the case [25]. Because medical equipment are now connected to the internet and possibly subject to cyber-attacks, the IoMT has raised several new security issues. Among the security methods recommended for IoMT devices include [26,27]:

Encryption: This entails encrypting all data exchanged between IoMT devices as well as other systems in order to avoid unauthorized users from intercepting it.

Authentication: This guarantees that only those with authorization have access to the IoMT devices and data. Multi-factor verification and other strong authentication systems can help prevent illegal access.

Access control: This entails establishing access control policies that limit authorized users’ exposure to IoMT equipment and information.

Secure boot: This guarantees that only trustworthy software may operate on IoMT devices by checking the technology’s authenticity during the initialization phase.

Firmware updates: Frequent firmware upgrades can aid in the resolution of vulnerabilities and faults in the software operating on IoMT devices.

Intrusion detection and prevention: This includes the use of tools and procedures to identify and prevent unwanted exposure to IoMT equipment and information.

Physical security: This includes safeguarding the physical environment in which IoMT devices are placed in order to protect them from physical threats.

Continuous monitoring: This entails regularly monitoring IoMT devices and their data for unusual activities and dealing with any discovered risks as soon as possible.

Network segmentation: Isolating IoMT devices from the main network reduces the system vulnerabilities and prevents deflections in the case of an intrusion.

Body interaction with the individual, on the other hand, is all that is required to obtain access to the key and to remove it. Specific varieties of assaults, including radio intrusions and impersonation strikes, are also sensitive to many home automation equipment [28]. Another problem with IoMT devices is that these frequently depend on a private interface for platform and transmission channel security. No other encryption techniques are used with authentication methods, leaving the communication routes between the sensors and the controllers exposed, as stated in the preceding section. This research demonstrates a data masking mechanism that can protect against these assaults. There are a number of lightweight frameworks available to aid with the security of IoMT devices. Here are a few such examples:

Arm Mbed TLS: This is a lightweight version of the Transport Layer Security (TLS) and Secure Sockets Layer (SSL) protocols that may be used to encrypt communications between IoMT devices as well as other platforms. It is designed for usage on low-resource devices and supports a wide range of cryptographic techniques.

WolfSSL: This is another lightweight TLS/SSL solution that is intended to be very portable and simple to incorporate into embedded devices. It supports a variety of cryptographic techniques and is freely accessible through open-source licensing.

RIOT OS: This is a lightweight operating system intended for use with resource-constrained IoT devices, such as IoMT devices. It has a number of security features, including as secure boot, encrypted communication systems, and cryptographic algorithm implementation.

These frameworks include encrypted communication protocols, compatibility for cryptographic methods, and secure boot processes, among other security features that can assist to improve the security of IoMT devices [29,30,31]. However, no framework can guarantee perfect security; thus, it is critical to thoroughly examine the security risks associated with IoMT devices and adopt suitable security measures to reduce these threats. Furthermore, in this experiment, the lightweight framework used is a simple machine learning process. Here, the encrypted cryptographic key is generated using a PUF which provides random numbers. Additionally, as the keys or any kind of data are not stored on server, it has the least chances of being breached.

The proposed model, the Octopus approach, employs PUFs, which create the cryptographic keys required for signal verification. Sensitive data components are replaced with an illegible value using masking. Since it is not a true encryption method, it is impossible to recover the original value from the disguised value. It employs a technique known as de-identification, which involves removing or masking personal identifiers such as the name and social security number as well as omitting or summarizing quasi-identifiers such as the date of birth and zip codes, or, in this instance, any information pertaining to healthcare. Data masking is therefore one of the most often used methods for live data anonymization [32]. PUFs allow intruders to be blocked to a great extent and system protection to be reconfigured if required. PUF-based verification systems of various types have been suggested for implementation in the IoT context. In the Internet of Things, device identification is a serious concern. There have been several cases of assaults on IoT networks using rogue devices in the network undermining protection [33]. The lower cost of safeguarding large data deployment is a vital advantage of this strategy. Masking lessens the requirement for implementing extra security controls on the data while they are stored in the platform as safe data are transferred from a secured origin into the system [34]. There are several advantages in healthcare data protection using data masking. For example, it has effective usage in clinical and healthcare industry trials, and analyzing these trials would be useful for future research. Another major advantage is that it can prevent death by protecting and securing healthcare data from attackers who can inject erroneous data. Moreover, data masking can help doctors to make urgent but wise clinical decisions in emergency conditions when the attacker is trying to breach the data to cause a critical situation. The main contributions of this experiment are as follows:PUF is used to mask health data. The classified data are used to construct a response, which is the masked data.A lightweight framework is created in which only simple and minimal processing is required.Generally, after the data collection, the data are processed using mathematical operations and encryption. There is a probability of attacks exploiting the data before they are even processed. Using the proposed methodology, the data are masked using PUF before any kind of data processing is initialized. This avoids attacks in the data collection stage.A simplified ML model is used to retrieve the original healthcare data with substantially less computational time, which makes this technique more attractive.Generally, a large dataset is used when the PUF is involved. In the proposed experiment, the ML model replaces the requirement of a large CRP dataset.The healthcare data are not sent in plaintext to the server, which makes the approach more secure and lightweight.In this research, timestamps are used as a part of the challenges of the PUF. Even if the health data are identical for a particular person at different timestamps, the masked data will be different for different timestamps.The proposed framework gives high accuracy, which shows that it can be used to retrieve the original health data.

## 3. Physically Unclonable Function (PUF)

The requirement to communicate data securely has always existed. However, due to the fast expansion of digital communications, this demand has expanded drastically over time [35]. Previously, all cryptographic elements were evaluated statistically, as if they were black boxes, with prospective attackers only seeing the input and output but not the underlying activity. However, such expectations have proven exceedingly difficult to achieve in practice, necessitating loss protection measures [36]. In terms of key preservation, keys are often kept in non-volatile memory to be used in algorithms. The aim of a PUF is to produce physical elements with distinct and unexpected behavior by using unpredictable material variances created during the production process [37]. When the same challenge is presented to another PUF version, even though it was built using the same procedure, the result is different. The definition and benefits of a PUF are outlined in full below.

### 3.1. Definition

A PUF is concerned with the study of things that demonstrate identifiability and physical unclonability and have a challenge–response capability. Public electronic medical equipment that are accessible by the patient are referred to as end devices. Based on the conditions, these devices are connected to a network. The information from the devices are delivered to cloud services over the internet, and the doctor has access to it via a local area network. All the WMDs have a PUF module planted in them. The production differences that arise during the creation of an integrated circuit are used in a PUF. Variations occur throughout the production process as a result of the procedures involved, and these are included in the systems, which distinguish them from one another. Because no two devices on a given wafer are alike, they generate various outputs. The PUF makes use of this unpredictability while generating cryptographic keys. Uncertain, uncontrolled, inevitable, and natural changes are inserted into the gadgets. As a result, the PUF device’s output cryptographic keys are likewise inherently randomized. PUFs generate a unique digital output depending on the device’s physical features. Variations in manufacturing, temperature, voltage, and other variables that impact the circuit’s behavior are examples of these features. Because each device has its own set of features, the output of a PUF is also unique and cannot be simply copied. PUFs are also designed to be resistant to many types of assault, such as side-channel attacks and fault injection attacks. To obtain the secret key, side-channel attacks require examining the physical properties of the device while it is executing cryptographic operations. PUFs, on the other hand, are designed to create random and unexpected output even in the face of such attacks. Overall, including PUF-generated input keys into ML models can give a high level of protection against threats. The input to a PUF is referred to as a “Challenge”, while the output is referred to as a “Response” [38], as shown in Figure 4.

The PUF’s input–output pair is known as a Challenge–Response Pair (CRP) and is often used to authenticate the gadget. For various wafers, the differences that occur throughout the manufacturing method are neither uniform nor consistent. Various PUF modules respond differently to the same challenge input. As a result, the PUF signal becomes a signature of that particular IC. The credentials are created without using any of the device’s primary processor’s computing capacity. As a result, the overall structure is both compact and safe. The credentials are also not kept in the sensor’s memory, making them immune to numerous side-channel assaults. Based on the cryptographic protocol or the security mechanisms used, credentials can be created as needed. This also cuts down on the amount of storage needed to enable this authentication mechanism in the environment of the IoMT. Based on the manufacturer and structure of the PUF, the number of data pairs from an individual PUF module might be extremely large. The architecture’s resilience is determined by the PUF module’s essential features [39], as shown in Figure 5. A controlled PUF is a PUF that has been integrated with control logic that restricts the possible evaluation methods. In general, the controlled PUF is locked without consent from a reliable source, and no answer can be effectively analyzed. More CRPs can be retrieved once a user has been granted access to one. Similar to key management, multiple session keys can be generated from a master key in this instance. The controlled PUF is constructed in practice such that the PUF and its control logic may perform complimentary functions [40,41].

### 3.2. Figures of Merit of a PUF

As a reliable and portable option to secure IoMT devices, PUFs are suggested by all researchers. The figures of merit of a PUF are explained below.

Reproducibility: Since the same challenge is presented to any PUF server, it is considered to be repeatable if the response would still be the same or nearly identical. Intra-distance is the closeness between two replies to the same challenge and PUF instance, and it is commonly assessed utilizing Hamming Distance (HD) [42].

Uniqueness: While the same challenge is issued to several PUF instances, their replies are significantly diverse; this is referred to as the uniqueness attribute of a PUF (i.e., their inter-distance is large). Inter-distances are frequently determined utilizing Hamming Distance in the same manner as repeatability. Uniqueness, on the other hand, is normally assessed under standardized operating settings and not under variable situations [43].

Identifiability: When PUF cases can be recognized by analyzing their answers, they have the identifiability attribute. The PUF must have the traits of repeatability and uniqueness to satisfy these objectives. It is worth noting that a PUF does not have to be flawless in terms of repeatability or originality in order to be recognized. In reality, designing PUFs with an aggregate intra-distance of 0% is nearly difficult. Intra-distances, on the other hand, follow a confidence interval centered on a minimal value. Similarly, average inter-distances often reflect a confidence interval centered on a value less than 50%. The criteria for being recognizable is that intra-distances should be less than inter-distances with a large gap [44].

Randomness: The PUF key’s unpredictability is the indicator of equilibrium between the number of ones and zeros in a PUF key. A safe PUF module that can withstand extreme strength, as well as other critical guessing assaults, is one in which the PUF key contains an equal number of randomized zeros and ones [45].

## 4. The Proposed Model for IoMT Device

This section presents the proposed Octopus model. Figure 6 shows the elements and overview of the proposed method. Health data are masked using the PUF on the patient’s device, and data are retrieved using an ML model in the cloud server (CS). All equipment that are introduced to the network have a PUF module built into them. The general system model consists of the IoMT device, a gateway, and the cloud server. Here, the IoMT device is a medical device, namely, a resource-constrained wearable gadget that is connected to the patient. The device gathers data about the patient’s condition and provides it to the doctor via the portal. In the proposed Octopus method, the WMD has a PUF module. Moreover, here, the gateway connects the device with the CS. Data from the patient’s device are transferred to the CS using a gateway. We presume that the gateway is a reliable source of information and that it has sufficient storage and processing power. Now, the CS is a centralized server used to store all the patient’s information. After obtaining the masked data from the medical device, the CS retrieves the original data using ML. To retrieve a patient’s health information, the doctor authenticates with the gateway using a resource-constrained smartphone or tablet. The doctor must first authenticate with the gateway in order to connect with the sensor network.

Table 1 presents the acronyms used in the method. The method is divided into two phases: (1) Enrollment phase, and (2) Data masking and retrieving phase.

### 4.1. Assumptions

The success of data masking and retrieval was predicated on the following hypotheses:The WMD is incorporated with PUF chips.The PUFs of medical devices are strong and unaffected by outside variables such as temperature, voltage, current, humidity, noise, etc.The ML model is only stored in the secure database (SDB) of the CS. Only the server can access the ML model to retrieve the health data.No CRPs are stored anywhere.Before data masking, the WMD is already verified in the network.

### 4.2. Machine Learning Algorithm

An ML classification problem with more than two classifications, or outputs, is known as multi-class classification. Multi-layer Perceptron (MLP) is a deep-learning-based ML model. Since each image may be classified in as many distinct animal categories as possible, using a model to identify animal species in photographs from an encyclopedia is an example of MLP. MLP also necessitates the use of just one category in a dataset. MLP is perhaps the most widely used ML application, aside from regression. We are given a set of training samples separated into K distinct classes, and we create an ML method to forecast which of those classifications some previously unknown information relate to. The model can learn characteristics unique to each class from the training dataset and utilizes those similarities to forecast the participation of the new dataset.

In the proposed framework, for the multi-class classification problem, we used multi-layer perceptron, which is a deep-learning-based ML model. In multi-class classification, the task is to classify input data into one of several possible classes. MLPs can be used to perform this classification by learning non-linear mapping between the input data and the corresponding class label. This algorithm was chosen because healthcare data can have a range of data and each datum can be leveled. Using MLP, it is possible to identify the level and predict the actual health data. For example, when an IoMT device is used, it gives out a reading that can range between X and Y. This reading can be leveled as 0 to ((Y − X) − 1) and this changed dataset can be trained to predict the actual IoMT device data. MLP makes the prediction that each sample is assigned to one and only one label.

The IoMT device has a PUF in it which collects the data from the human body and the timestamp. The data and timestamp act as challenges of the PUF, which will generate responses. An attacker would be unable to train a comparable model if the PUF-generated output is utilized as input keys for an ML model since they would not have access to the same unique input keys used to construct the original model. Even if the attacker gained access to the PUF-generated output, they would be unable to recreate the original input keys since the output is produced by a physical process that cannot be manipulated or replicated. The responses along with the challenges (health data and timestamp) are sent to the cloud server. The server then prepares a dataset using those data where timestamps and responses are the input features and the IoMT device’s data are the output feature. Here, ML is used for predicting and retrieving the masked health data from the IoMT device. MLP is used in this paper because healthcare data are sequential data that are used in a particular range. The data are not in a discrete range such as housing prices; hence, it can be leveled at each point of health data. Therefore, MLP is a more suitable method than linear regression or any other algorithm.

### 4.3. Enrollment Phase

Before a device latches to a network, it is required to be enrolled on the server. The enrollment phase is divided into two steps. Figure 7 shows the process of enrolling the WMD.


PUF response generation: As shown in Algorithm 1, initially, DT and $T_{st}$ are selected, and the combination of these acts as challenge *C* of the PUF of the medical device. The PUF then generates a response *R* using the process variation of the chip. The response, timestamp and health data are then shared with the CS through a secure communication medium.$ML_{model}$ Training and database storage: In this step, an ML model is trained using the received *R*, DT, and $T_{st}$. The server uses *R* and $T_{st}$ as the input features and DT as the output feature. The generated model is then stored in a SDB for data retrieval. This completes the enrollment phase.


**Algorithm 1:** Secure Enrollment Process

**Step-1: PUF response generation**
WMD:   DT
||
$T_{st}$ = *C*   *C* → *R*WMD ⟶ CS {*R*, $T_{st}$, DT }
**Step-2: $ML_{model}$ Training and database storage**
CS:   *R*, $T_{st}$, DT ⊧ $ML_{model}$CS ⟶ SDB {$ML_{model}$}SDB:   ∈ $ML_{model}$


### 4.4. Data Masking and Retrieving Phase

The proposed method is presented in Figure 8. The developed scheme is divided into two steps, as shown in Algorithm 2:Data Masking: Data are masked using the incorporated PUF in the WMD. First, the WMD collects DT from the human body. Moreover, $T_{st}$ is identified from the clock of the WMD. Before doing any kind of further operation, both DT and $T_{st}$ act as challenge *C* of the PUF, which will generate response *R*. This response *R* is then sent to the CS using a public channel.Retrieving Data: Upon receiving *R*, CS selects $T_{st}$ and uses both *R* and $T_{st}$ as the input features of the stored $ML_{model}$ for that WMD. The $ML_{model}$ then predicts the actual data DT. By following this method, CS can retrieve the masked data from the WMD.
**Algorithm 2:** Data Masking and Retrieving Process**Step-1: Data Masking**WMD:   ‡ DT, $T_{st}$   DT
||
$T_{st}$ = *C*   *C* → *R*WMD ⟶ CS {*R* }**Step-2: Retrieving Data**CS ⟶ SDB {ID}SDB:   ID ∋ $ML_{model}$SDB ⟶ CS {$ML_{model}$}CS:   ‡ $T_{st}$   *R*, $T_{st}$ ↦ DTCS ⟶ SDB {DT}SDB:   ∈ DT

## 5. Experimental Results

The functionality of the suggested authentication mechanism is examined in this section. Moreover, the experimental setup, dataset preparation, and ML model training and prediction are also explained in this section.

### 5.1. Experimental Setup

In order to avoid the difficulties posed by the problem of latency in data processing, this project utilized the Jupyter Notebook. Furthermore, the computer setup had a 3.40 GHz Intel Xeon processor, 48 GB RAM, and NVIDIA RTX A4000 32 GB GPU. The model was created and tested on a Jupyter notebook because it comes with built-in support for GPU-enabled TensorFlow and the requisite CUDA acceleration. This was decided with an eye toward the model’s ease of replication by the research community. There are numerous PUF architectures that can produce CRPs with the necessary properties. As shown in Figure 9, the component used in this setup was an IoMT device on the client side. Raspberry pi was used as an IoMT device and the server in this experiment. Moreover, an ML model was saved on the server to retrieve the data. The other component used in this research was the 64-bit arbiter PUF. The PUF was constructed using the Xilinx BASYS3 FPGA. A delay-based PUF that creates a signal based on the difference in the times of two delay lines is called an arbitrator PUF. The challenges are gathered based on a Xilinx BASYS3 FPGA’s 64-bit PUF implementation. A delay-based PUF that creates a response based on the difference in the times of two delay lines is called an arbitrator PUF. The changes in the micro-electronic production process that cause a race between two identical pathways are the basis for how this PUF functions. The race has an impact on the value that the arbiter latches and is related to the variation in signal propagation latency on these two channels. The only relevant aspect of this difference is its sign, not its precise value. The sign that the arbitrator derived represents the responses and serves as the PUF identity. The arbiter can be built as a straightforward SR-latch using two cross-coupled NOR gates.

Figure 10 shows that the unit of two delay lines to create a bit is represented by each box between A0 to A63. A series of red and blue colored boxes (2 × 1 multiplexers) is seen within each box, designating two distinct lines for processing signals. Every bit challenge serves as a multiplexer pair’s selection bit. The very first box between A0 to A63 includes challenge C0, the second box between A0 to A63 includes challenge C1, and so on. When one transmission is delivered in the PUF, the signal flows via the multiplexers in accordance with the challenge’s selection bit. For instance, if C0 is 0, the transmission will flow to multiplexer C0, and if C1 is 1, the transmission will travel line 1 of the following pair of multiplexers (between red and blue). In the end, every box of the multiplexers is linked to a D flip-flop. If the flip-flop’s D input receives the signal more quickly, its output indicates that the response bit would be 1. This method uses multiplexer pairs (from A0 to A63) to convert a 64-bit challenge (from C0 to C63) into a 64-bit response (from R0 to R63).

### 5.2. Dataset Preparation

As shown in Figure 11, 64 bits were used for generating the challenges. All possible human body sample temperature values were generated for an entire year. The temperature was noted at every 5 min interval for the whole year. The first 8 bits were used as the temperature, which are 95 to 105, and the next 8 bits were the temperature after the decimal, which are from 0.0 to 0.9. In this way, the temperature was divided into 16 bits. The next 16 bits were allocated to the months, so the first 8 bits were the first digit of month which is just 0 and 1, and the next 8 bits were for the second digit of the month, which is from 0.0 to 0.9. This is how the months were separated into 16 bits. The next 8 bits were for the first digit of the dates used, which is 0 to 3 and the other 8 bits were the second digit of the dates. For the next 8 bits, the hours in a day were used, which is 0 to 23, and for the last 8 bits, the minutes in an hour were used, which is divided by 5. So, this way, the last 8 bits were from 0 to 11. For this, the challenges were created using Python programming. In this experiment, responses were generated using a Xilinx BASYS3 FPGA’s 64-bit arbiter PUF following challenges from a Raspberry pi. The challenges and the responses were merged to make a 128-bit CRP. A total of 11554460 CRPs were generated as the dataset. These CRPs were later converted into binary and were used in ML models to train the model and obtain the best accuracy. The first 16 bits are considered as the output feature and the remaining 112 bits are considered as the input feature. Similarly, the blood pressure dataset was created where the first 8 bits were the blood pressure data, which was in the range of 40–200. The next 8 bits were ’0’. The remaining were similar to temperature dataset. Just like the temperature dataset, this dataset was converted into binary and was used in ML models to train and obtain the best accuracy.

### 5.3. Computation Cost

The suggested framework is for IoMT equipment that have limited resources. The sophistication or duration spent in activities increases the stress on such equipment. The actions performed during authentication are minimal and do not consume a lot of resources. The standard time required for the PUF to generate the responses vary depending on the specific implementation and type of PUF used. PUF response generation, in general, entails sending a challenge signal to the PUF and evaluating the response. The calculation time required for this operation is determined by factors such as the complexity of the PUF circuitry, the measuring equipment’s speed, and the nature of the challenge signal utilized. When a challenge was sent, the PUF delivered a response in 0.4 μs, having a minor influence on the proposed protocol’s calculation time [46]. Moreover, the encryption and decryption time for 256 bits varies based on the encryption and decryption technique, mode of operating, and equipment. The Advanced Encryption Standard (AES) is a popular encryption technique that may employ 256-bit keys. With contemporary technology, AES-256 encryption can encrypt tiny quantities of data in a matter of milliseconds. Nevertheless, for greater volumes of data or if the encryption is conducted on less capable hardware, the encryption time might grow dramatically. An AES Encryption 80-bit message with 256 bits has a computation time of 0.23 ms, whereas the decryption time is 0.14 ms [47]. The amount of energy spent during data encryption varies greatly depending on a number of factors, including the exact PUF design, the encryption technique employed, and the quantity of the data being encrypted. The comparison of energy consumption depending on different factors is shown in Table 2. This shows that the total energy consumed to encrypt data might range from a few hundred microjoules to several millijoules [48]. Moreover, in the presented framework, the communication overhead is 64 bits, whereas all the other frameworks used have a communication overhead of 128 bits and above, which makes our proposed framework different from usual. Furthermore, the energy consumption of the arbiter PUF used here is about 2.2 mW to 12.5 mW. Comparing all these factors, our proposed approach can provide the same level of security feature with less overhead and energy.

### 5.4. Machine Learning Model Training

We used the Jupyter Notebook to train and test the machine learning model. In this experiment, a deep-learning-based ML model (MLP) was used. A number of feedforward, deep architectures were used for model training and evaluation. The dataset was first divided into two parts. Eighty percent of the dataset was used to train the model, and the remaining twenty percent was used for validation. Categorical data was fed because the dataset had more than one discrete item, namely, the temperature, which was needed as the output feature. The ML model was divided into two parts: The first part was to train the dataset for the temperature before the decimal, which is 95 to 105. The latter part of the model was used to train the temperature after the decimal, which is 0.0 to 0.9. Here, temperature was considered as the output feature which was 16-bit, and the rest were considered as the input features. The training data, testing data, and all the ML-based setups can be found in Ref. [51].

As shown in Table 3, many models were used for this experiment. A deep-learning-based ML model, namely, MLP was used in this experiment to retrieve the original data. This way, the model preserves the security and privacy of the data and the user. Here, if the temperature of the patient changes, the model itself is able to retrieve the original or changed data. In the first model, 4096-4096-4096 layers were used with no dropout. Here, batch normalization was used, and the optimizer used was “Adadelta” with a total of 25 epochs and a batch size of 10,000. Here, “Swish” was the activation function that was used. The validation accuracy for this model was 94.23%.

For the next model, it used 4096-3072 layers without any dropout. The optimizer utilized here was “Adam” with a total of 10 epochs, and the batch size was 5000. Instead, batch normalization was applied and the validation accuracy here was 86.08%. The following model had a 30% dropout rate and used 4096-4096-4096-3072-3072-3072-2048-2048-2048 layers. The batch size was 10,000, and the optimizer used was “Nadam” with a total of 50 epochs. Batch normalization was used here to obtain 95.01% validation accuracy. All other models can be found in Table 3.

As the temperature series is from 95 to 105, there are 11 classes. Our first ML model was comprised of four hidden layers, with ($2\times 2^{n}$) neurons for the first and second hidden layer. The third hidden layer here was comprised of ($3\times 2^{n}/2$) neurons. The fourth hidden layer was made of ($2^{n}$) neurons, as shown in Figure 12. As the total number of classes is 11, the value of n is 11.

For the model used in this experiment, 4096-4096-3072-2048 was the combination used with the layers of batch normalization. The activation function here was “Swish”. There was no dropout in this model. The optimizer used in this model was “Adam”. “Categorical Crossentropy” was used as the loss function because there were more than two output labels. Ten epochs in total were used, with a batch size of 5000. The metric here is “accuracy” as the models need to predict the output feature. As shown in Figure 13, 96.07% is the validation accuracy of this ML model, which was used for the first part to obtain the output feature, namely, the temperature before the decimal.

Similarly, the second model was comprised of four hidden layers with ($4\times 2^{n}$) neurons for all the four hidden layers. Moreover, as the decimals here were from 0 to 9, the number of classes is 10 and the value of n is 10. The combination utilized with the layers of batch normalization was 4096-4096-4096-4096. Here, the “Swish” activation function was used. This model did not have any dropouts. “Adamax” was the optimizer employed in this model. Given that there were more than two output labels, the loss function was “categorical crossentropy”. There was a total of 10 epochs used, with a batch size of 10,000. Since the models must be able to predict the output feature, “accuracy” is the key metric here. This ML model was employed for the second part’s output feature, namely, the temperature after the decimal, and it obtained a validation accuracy of 89.83%, as shown in Figure 14.

Similarly to the temperature health data, an ML model was created to train and test the blood pressure data. In this model, the dataset was divided into two parts as well. The model was trained using 80% of the dataset and validated with the remaining 20%. Because the dataset had more than one discrete item, namely, the blood pressure data, which was required as the output feature, categorical data was given. In this case, blood pressure data was deemed as the output feature, which was 8-bit, while everything else was considered the input feature.

Several models were utilized in this experiment, as stated in Table 4. Just like the previous model, a deep-learning-based ML model with an MLP was employed to recover the original data. With this approach, the model protects the security and confidentiality of data as well as the user. If the patient’s blood pressure changes, the model will be able to obtain the original or altered data. The very first model used had 4096-3072-2048 layers with no dropout. The optimizer used here was “RMSprop” with 15 epochs and a batch size of 10,000. Here, batch normalization was used and the validation accuracy here was 92.82%. In the second model, 4096-3072-3072-2048 layers was employed without dropouts, which dropped the accuracy. Additionally, instead of “Adamax” or “Adam”, we again used “RMSProp” as our optimizer here. In contrast to the first model, we increased the epoch counts to 50 and the batch size to 5000. Due to these modifications, the validation accuracy was 90.15%.

Similar to the previous model, the number of classes is 17 since the blood pressure series ranges from 40 to 200. Our model for machine learning was made up of four hidden layers, each having ($4\times 2^{n}$) neurons. Because there are 17 classes in all, the value of n is 17. The used sequence was 4096-4096-4096-4096. The “Relu” activation function was employed in this case. There were no dropouts in this model. In the ML model used here, the optimizer “Adam” was used. A total of 30 epochs are employed, with a batch size of 10,000. Because the models must be able to predict the output feature, “accuracy” is the most important statistic. This ML model is used for the blood pressure data output feature and has a validation accuracy of 97.42%, as shown in Figure 15.

### 5.5. Temperature Prediction

In this experiment, the maximum accuracy found for model 1 was 96.07%, whereas 89.83% was the maximum accuracy found for model 2. To obtain better accuracy, a different kind of method was implemented. For this part of the implementation, 1,000,000 CRPs were used as the testing data. As shown in Figure 16, there were four scenarios. In all the scenarios, there were three different time stamps considered. The actual value 1 and actual value 2 are the temperatures that were selected in sequence with an interval of time. The predicted value 1 and predicted value 2 are the values that were predicted using the ML model. Moreover, the pre-accuracy is counted as “1” only if the actual value is matched with the predicted values; otherwise, the pre-accuracy is counted as “0”. If the pre-accuracy value is 2 or more, the accuracy will be counted as “1”; otherwise, the accuracy will be counted as “0”.

In the first scenario, none of the actual values matched the predicted values. Hence, the pre-accuracy was counted as “0” and the accuracy was “0”. In the second scenario, the very first actual value and the first predicted value were in agreement, but the remaining two actual values and predicted values were not. The pre-accuracy was therefore recorded as “1” for the first-time stamp and “0” for the following two. Additionally, accuracy was “0” as a result. The first and second actual values as well as the first and second predicted values were in agreement in the third scenario; however, the last actual values and predicted values were not. As a result, the pre-accuracy was noted as “1” for the first and second timestamps and “0” for the final one. As a result, accuracy was marked as “1”. In the fourth scenario, each of the actual values and predicted values were exact matches. As a result, the accuracy was “1” and the pre-accuracy was “1” for each of the time stamps. A total of 1,000,000 CRPs were tested using this method and the accuracy after this implementation was found to be 99.45%. The result for this experiment is shown in Figure 17. Similarly, the same technique was used in the blood pressure prediction experiment. Likewise, 1,000,000 CRPs were tested using this approach, and the accuracy after execution was 99.64%. Figure 18 presents the outcome of this experiment.

### 5.6. Security Analysis

This section discusses how the suggested framework handled several security problems. It also highlights how the established authentication system may apply security measures while taking into consideration an adversary’s ability to change and listen in on data transferred over public networks. These are as follows:

Side-Channel Attacks: Side-channel attacks are often carried out by monitoring computation time, power analysis, and so on. Furthermore, encrypted keys kept in the device’s memory increase the likelihood of being impacted by side-channel attacks. In order to avoid this, responses were generated using a PUF. The data were not saved in the device’s memory or utilized further by the medical device in the suggested technique. Hence, the shown method can be used to avoid side-channel attacks.

Physical attacks: Physical attacks can be carried out by gaining access to the encrypted keys stored in the device’s memory. If an attacker attempts to interfere with the PUF-based device, the device will be harmed and will provide inaccurate replies. Furthermore, neither the challenges nor the replies are saved in the devices. As a result, physical attacks have no effect on the framework.

Modeling Attacks: In modeling attacks, an adversary attempts to capture the pattern of secret keys in order to construct a model that predicts the device’s next keys/responses in order to disrupt the system. It is feasible to defend the CRP interface from modeling assaults. It is also possible to hide the interface by supplying an extra block, hash function, etc. The proposed solution took advantage of PUF to provide a partial response that was completed by the device itself, and the challenge was not sent. Furthermore, the created answer was hidden before being sent to the CS. As a result, the suggested technique can withstand modeling attacks.

Eavesdropping Attack: Because each PUF has unique properties, the CRP assures that only the authentic device and server can communicate with each other. Eavesdroppers cannot reproduce it without access to the model and CRPs since the server only keeps the model during the enrollment procedure and has no knowledge about the responses.

Forward Secrecy:The fundamental purpose of forward secrecy is to ensure that previously established session keys remain secure even if the keys are hacked. The proposed framework does not utilize any pre-stored keys to authenticate themselves. Furthermore, the response is disguised with PUFs, which are generated at random. As a result, without the random numbers, the secret key cannot be identified. Furthermore, different challenges, responses, data, and so on will be utilized for the next authentication and data masking, thus preserving forward secrecy.

The discussion has shown that the proposed architecture is resistant to recognized security concerns.

## 6. Conclusions and Future Work

Many areas of industry have been transformed by the Internet of Things. One of the first industries to seize this opportunity by populating the internet with medical-related items is the healthcare sector. Security has emerged as the main concern as a result of rapid growth and diversity. In order to find security flaws ranging from device assaults to data transit attacks, attackers and the research community are continuously focused on the creation and rapid expansion of IoT applications. Safeguarding IoMT devices has become crucial due to the demand for using IoMT sensors to lower healthcare costs and provide better care for patients. The main issue when using networks on a wide scale is security. The confidentiality and privacy of the patients is the main area where IoMT enables smart healthcare services. In this regard, verification and permission procedures are key security requirements since they ensure that sensitive medical data cannot be intercepted. In order to overcome the constrained hardware on cloud servers, the fundamental form of a distributed ML system with embedded devices was taken into consideration. To address the issue of data stealing in distributed ML systems, attempts were made to build a data-concealing framework. The experiment offered can demonstrably substantiate the validity of the data-concealing principle put out in this work. In the distributed ML system, the neural-network-based model can effectively fulfill the encryption and decryption duties from the perspectives of operability and being hard to steal. This is in contrast to the classic encryption technique. This paper presented a methodology that combines a mix of these strategies to achieve all security criteria, since no current strategy can completely satisfy the security requirements of these systems while also mitigating the majority of threats. Starting with data collection and ending with data storage and sharing, this architecture covers every stage of data and device security. Future directions and views in this area revolve around finding practical privacy and security solutions in the age of large healthcare data. In the proposed methodology, the accuracy that was found is 99.45% using the ML model. In the future, different federated learning models can be used to secure healthcare data. Furthermore, blockchain can be introduced to secure healthcare data. Other methods for protecting privacy must also be improved.

## Figures and Tables

**Figure 1 sensors-23-04082-f001:**
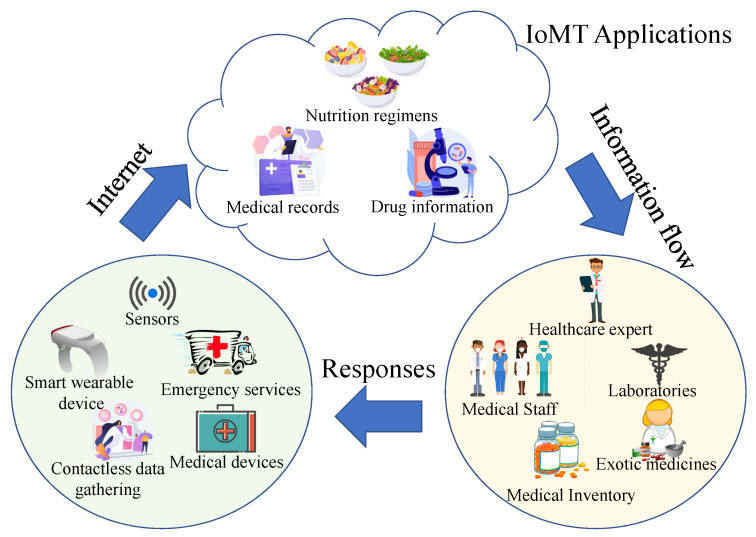
Internet of Medical Things and its applications.

**Figure 2 sensors-23-04082-f002:**
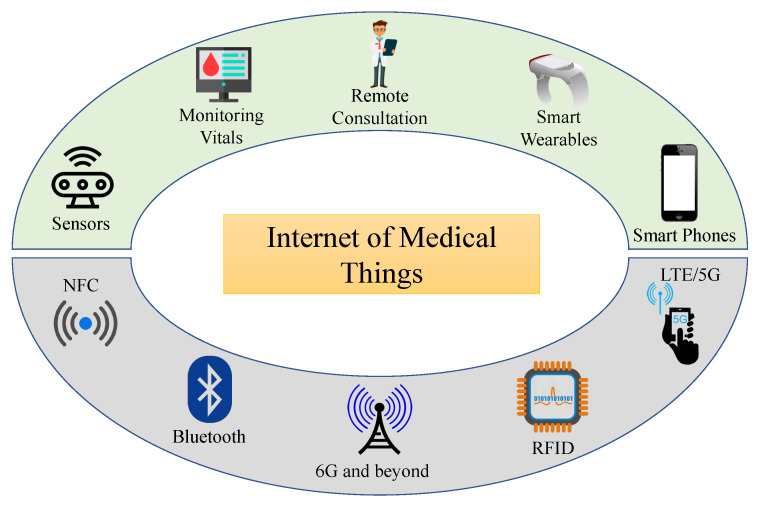
Internet of Medical Things enabling technologies and devices.

**Figure 3 sensors-23-04082-f003:**
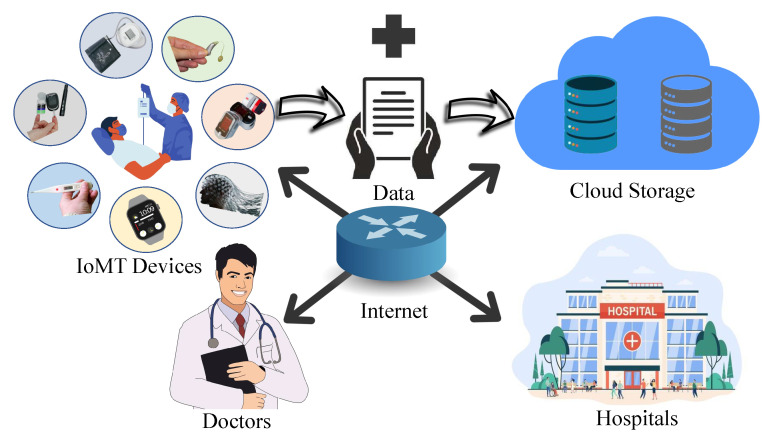
Security and privacy of IoMT devices.

**Figure 4 sensors-23-04082-f004:**
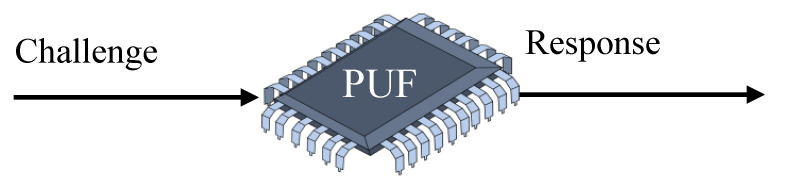
Challenge–Response Pair.

**Figure 5 sensors-23-04082-f005:**
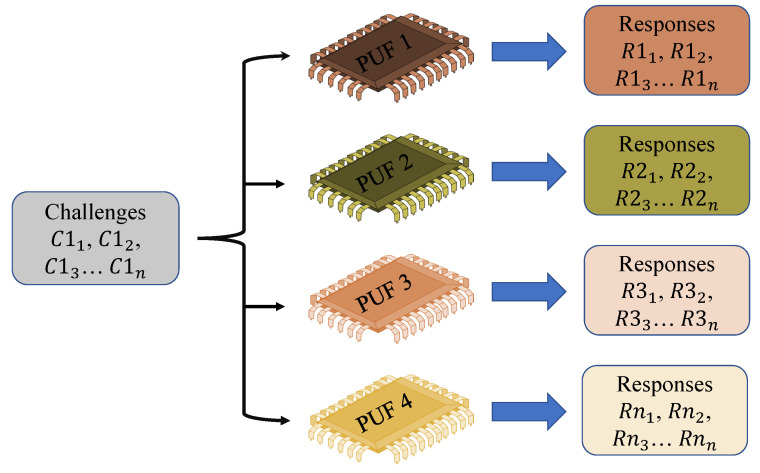
CRP preparation.

**Figure 6 sensors-23-04082-f006:**
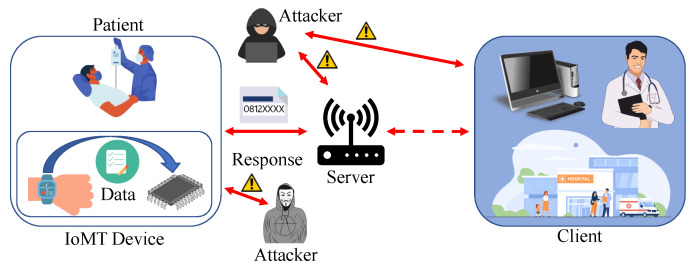
Overview of the Proposed Method.

**Figure 7 sensors-23-04082-f007:**
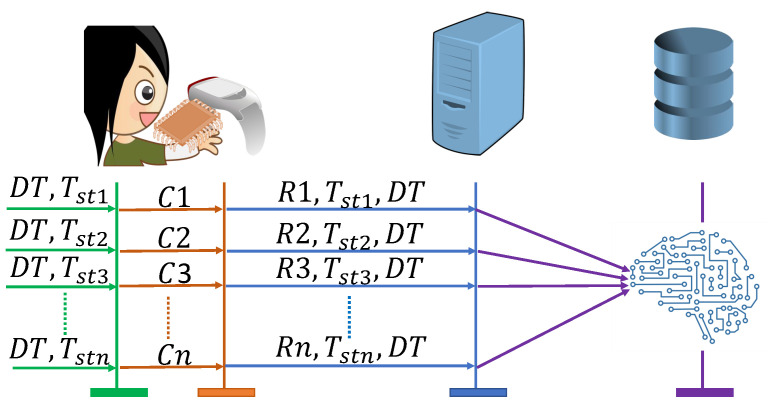
Enrollment Process.

**Figure 8 sensors-23-04082-f008:**
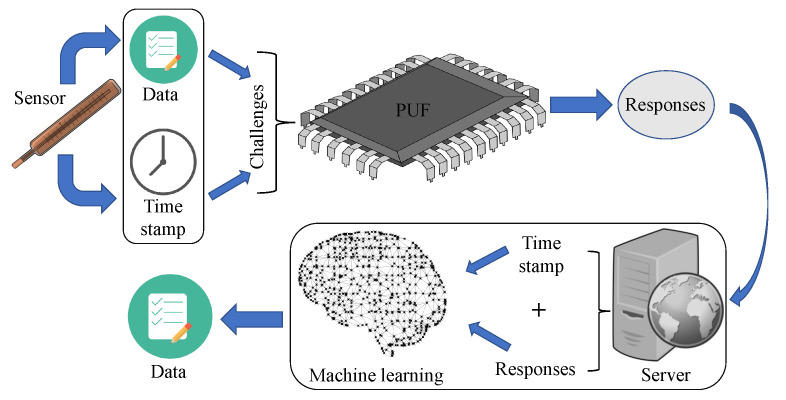
Proposed Model.

**Figure 9 sensors-23-04082-f009:**
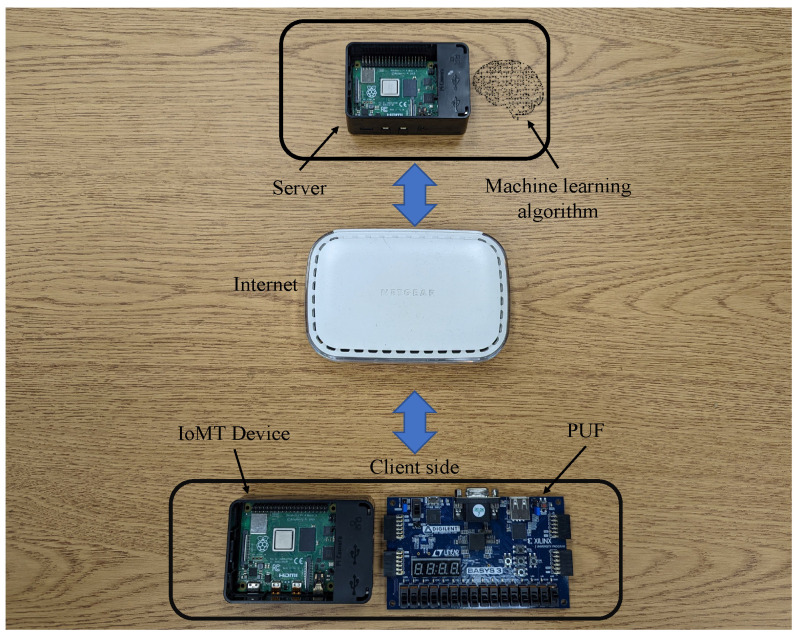
Experimental setup.

**Figure 10 sensors-23-04082-f010:**
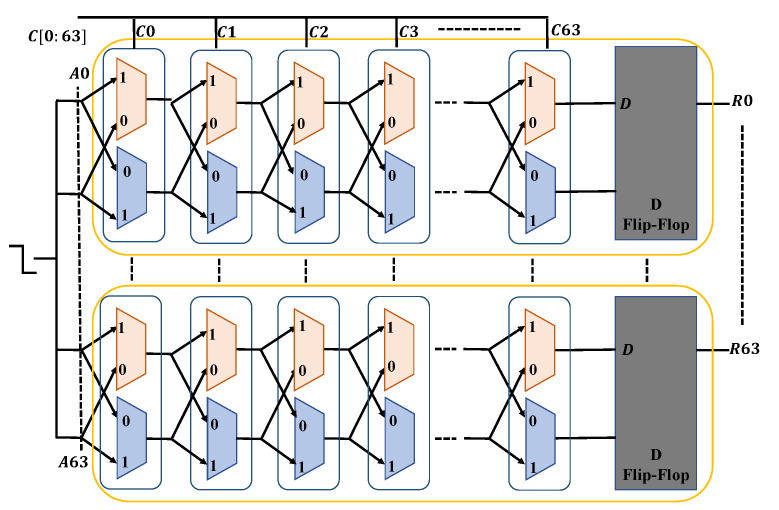
Arbiter PUF.

**Figure 11 sensors-23-04082-f011:**
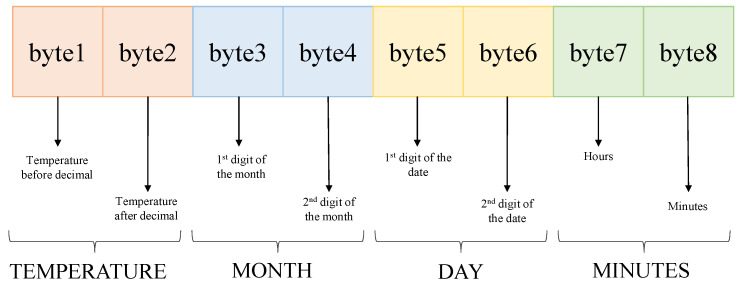
Data preparation format.

**Figure 12 sensors-23-04082-f012:**
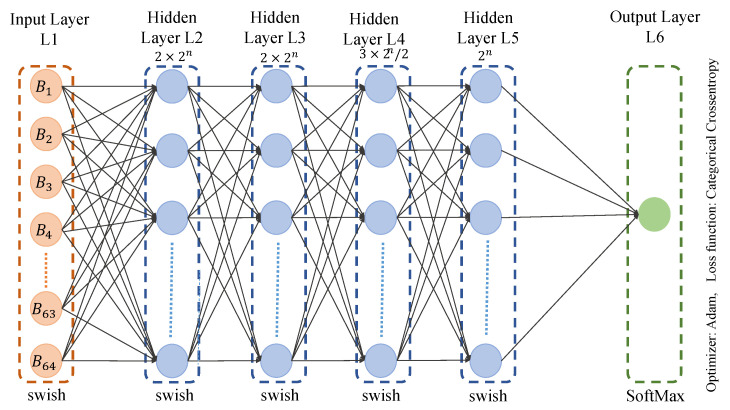
Multi-layer Perceptron architecture.

**Figure 13 sensors-23-04082-f013:**
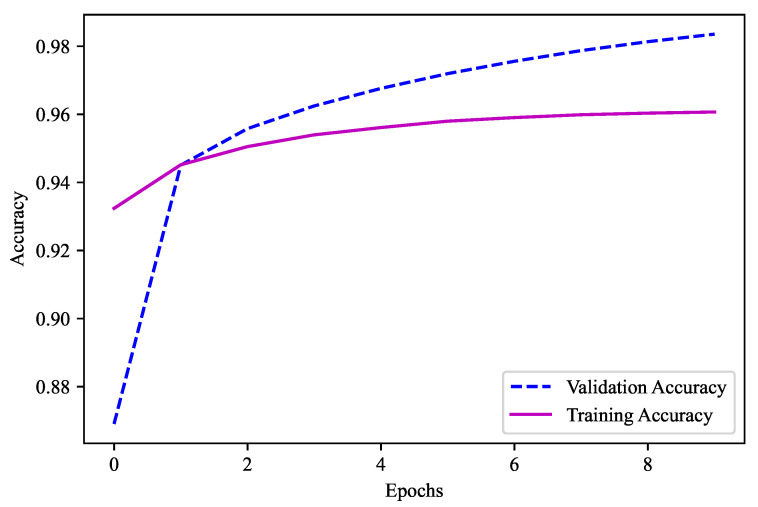
Validation accuracy for model 1 (Temperature 95–105).

**Figure 14 sensors-23-04082-f014:**
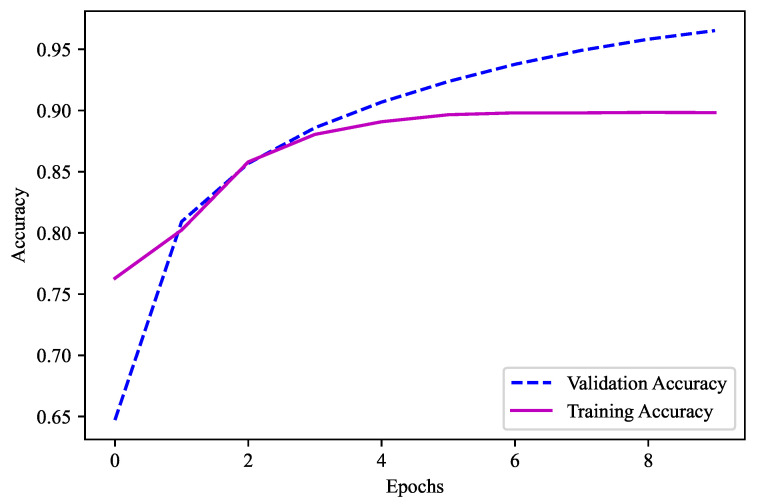
Validation accuracy for model 2 (Temperature 0.0–0.9).

**Figure 15 sensors-23-04082-f015:**
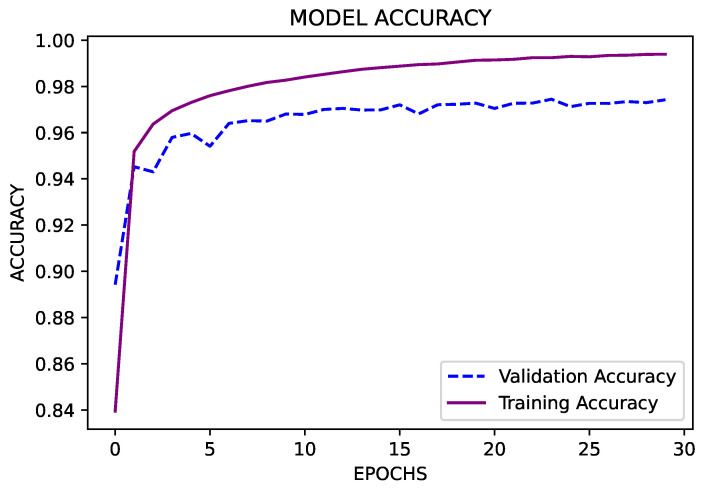
Validation accuracy for blood pressure data.

**Figure 16 sensors-23-04082-f016:**
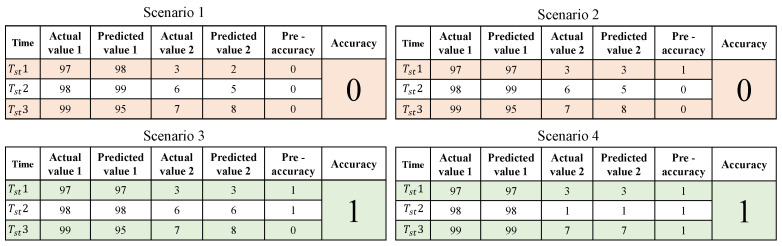
Temperature and blood pressure prediction algorithm.

**Figure 17 sensors-23-04082-f017:**
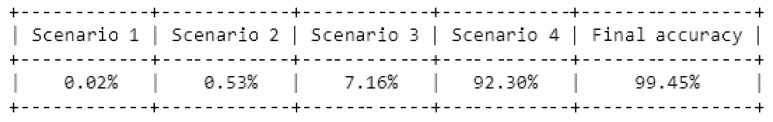
Experimental final accuracy for temperature data.

**Figure 18 sensors-23-04082-f018:**
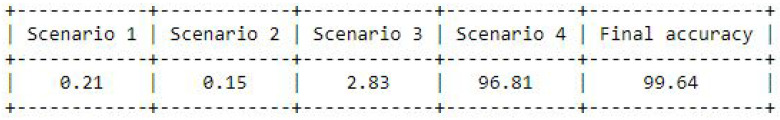
Experimental final accuracy for blood pressure data.

**Table 1 sensors-23-04082-t001:** Acronyms and symbols used in this paper.

Notation	Description
*C*, C1, C2, C3, Cn	Challenge
*R*, R1, R2, R3, Rn	Response
$T_{st}$, $T_{st1}$, $T_{st2}$, $T_{st3}$, $T_{stn}$	Timestamp
→	CRP Generation
‡	Data Collection
⟶	Data transfer
↦	ML model prediction
⊧	Model Training
∋	Database Query
∈	Store Operation

**Table 2 sensors-23-04082-t002:** Comparison of energy consumption [49,50].

Algorithm	Processor	Data Size	Energy Consumption for Encryption	Energy Consumption for Decryption
AES-128	ARM Cortex A8	1 KB	0.18–0.36 mJ	0.09–0.18 mJ
AES-256	ARM Cortex A8	1 KB	0.36–0.72 mJ	0.18–0.36 mJ
ChaCha20	ARM Cortex A8	1 KB	0.09–0.18 mJ	0.08–0.16 mJ
PUF	FPGA	—	2.2 mW to 12.5 mW	2.2 mW to 12.5 mW

**Table 3 sensors-23-04082-t003:** Comparison of a few models which were used for the temperature experiment.

Units	Dropout (30%)	Batch Normalization	Optimizer	Activation Function	Epochs	Batch Size	Validation Accuracy
4096-4096-4096	**✗**	**🗸**	Adadelta	Swish	25	10,000	94.23
4096-4096-4096-3072-3072-3072-2048-2048-2048	**🗸**	**🗸**	Nadam	Swish	50	10,000	95.01
4096-3072-3072-2048-2048-2048	**🗸**	**✗**	RMSProp	Relu	50	5000	91.05
4096-4096-3072	**🗸**	**🗸**	RMSProp	Swish	50	10,000	92.05

**Table 4 sensors-23-04082-t004:** Comparison of a few models which were used for the blood pressure experiment.

Units	Dropout (30%)	Batch Normalization	Optimizer	Activation Function	Epochs	Batch Size	Validation Accuracy
4096-3072-2048	**✗**	**🗸**	RMSprop	Swish	15	10,000	92.82
4096-3072-3072-2048	**🗸**	**🗸**	RMSProp	Relu	50	5000	90.15
4096-4096-3072	**🗸**	**🗸**	RMSProp	Swish	50	10,000	91.28

## Data Availability

Not applicable.

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
