# Peer review of "Octopus: A Novel Approach for Health Data Masking and Retrieving Using Physical Unclonable Functions and Machine Learning"

_sensors, 2023, doi:10.3390/s23084082_

Round 1

Reviewer 1 Report (New Reviewer)

The paper proposes using Physically Unclonable Functions (PUF) for secure health data storage and retrieval. The masked data is claimed to be retrieved using machine learning models. In my opinion, the manuscript needs significant improvements before it can be considered for publication. Although the paper aims to address security in the Internet of Medical Things (IoMT), no detailed security analyses of the proposed technique are provided to support this claim - this should be one of the main criteria for analyzing the proposed method. This is one of the main reasons that contributed towards my decision.

Some other comments for further improvements on the paper are listed below:

1. The introduction and problem statement can be more focused, highlighting related security mechanisms proposed for IoT/IoMT in the existing literature and the corresponding research gap.

2. The paper should have detailed security analyses concerning the different security goals. It is not clear from the paper whether the proposed method only achieves authentication with the usage of the PUF or if other security goals, e.g., confidentiality and integrity, could also be achieved.

3. If the proposed method claims to achieve confidentiality, how is it achieved? Based on Figure 7, the data DT is transmitted in plaintext to the server.

4. Based on Figure 8, assuming that the challenge input to a PUF resulted in the response, the client only sends the response to the server. And the data is then recovered by the server using a machine learning model? If that is the case, is there no secret (e.g., secret key) involved between the user and the server device? Have any analyses/proofs been conducted/provided on the claimed security of the transmitted data? Detailed analyses should include.

5. Is it reasonable to assume that the machine learning model is secret? Can an adversary build/train a similar model to retrieve the data or apply machine learning-based attacks (see the references below)? More clarifications/analyses are required towards this direction to support the author's security claims.

§  Hospodar, Gabriel, Roel Maes, and Ingrid Verbauwhede. "Machine learning attacks on 65nm Arbiter PUFs: Accurate modeling poses strict bounds on usability." In 2012 IEEE international workshop on Information forensics and security (WIFS), pp. 37-42. IEEE, 2012.

§  Tobisch, Johannes, and Georg T. Becker. "On the scaling of machine learning attacks on PUFs with application to noise bifurcation." In Radio Frequency Identification: 11th International Workshop, RFIDsec 2015, New York, NY, USA, June 23-24, 2015, Revised Selected Papers 11, pp. 17-31. Springer International Publishing, 2015.

6. From the performance analysis perspective, the authors presented their analyses on the accuracy of the models, which is acceptable. However, as the authors claim “Creating a lightweight framework” as one of the contributions of the paper, some indicators of energy/resource efficiency should also be discussed and compared against existing research to justify this claim. This would be useful as the IoT/IoMT devices are resource constrained.

7. On page 5, line 147, it is stated that “by flashing ultraviolet light on the individual, an attacker can acquire access to the code.” Is there any reference to support this claim?

8. It is recommended that the author publishes the experimental data/training data/code/setup via a public repository, such as Github. 

9. Some grammatical/sentence construction errors are observed. For example: in the first sentence of the abstract, "health equipment are…, on page 2, line 50 - "…. identification to a network is that its guarantees that…". These examples are not an exhaustive list. I recommended proofreading the paper carefully.

10. Suggest not to use TV shows (line 43 on page 2) as a reference.

11. Suggest avoiding contractions, e.g., it's, doesn't.

Author Response

Thank you so much for your valuable time. Please see the attachment, we have added point-by-point responses to all the comments.

Reviewer 2 Report (New Reviewer)

The proposed model used PUFs and create the cryptographic keys required for signal verification. Sensitive data components are replaced with an illegible value using masking. All necessary tests were made on the proposed model. The results showed that the obtained results of the proposed model are better than traditional methods. The paper has a high potential contribution to the literature.

I have some concerns:

1- There are unnecessary sub-sections in Sect. 3.2 and Sect. 4 (4.0.1 etc). They may be given by paragraph, instead of Sub-sections.

2- There are some unnecessary results in Tables 1 and 2. Only the best results and the parameter space of the DL model is enough to read the paper clearly.

3-  “multi-class classification which is a deep learning-based ML model”:  multi-class classification is the name of the classification problem for more than two classes. There is not any algorithm or classifier with the “multi-class classification” name. Also, the given architecture is not an ML method. If there is not any feature extraction and there are different layers, it is Deep Learning. The author must check the model again and clearly mention the names. 

Author Response

Thank you so much for your valuable time. Please see the attachment, we have added point-by-point responses to all the comments.

Round 2

Reviewer 1 Report (New Reviewer)

I think comment 2 of the previous review is not fully addressed. To address this comment, the authors seem to have included only about the details of desired security goals (and some details about security primitives/techniques) under Section 2. However, it is not clear whether the authors are claiming to achieve all these security goals in their proposed model (and analyses/proofs of the claimed goals).

Since the paper aims to address security in the Internet of Medical Things (IoMT), security analyses of the proposed technique should be one of the main criteria for analyzing the proposed method. I suggest adding a section on "Security Analyses of the Proposed Scheme (or something like that)" to provide detailed security proofs/analyses etc (perhaps under a subsection of Section 5) of the implemented/developed scheme.

Author Response

Thank you so much for reviewing the manuscript and giving your precious time. You can find the replies to your comments in the attached file here. 

This manuscript is a resubmission of an earlier submission. The following is a list of the peer review reports and author responses from that submission.

Round 1

Reviewer 1 Report

Authors have prepared an article related data masking for health data. I would suggest following points to authors,

1. What do you mean by Octopus? Why are you naming the proposed model as octopus?

2. You have mentioned that machine learning is used for retrieving data. There are no details of machine learning apart from using the term just to make the paper belong machine learning domain. Which algorithm is used? Infact, your problems statement is related security and data privacy, why did you use machine learning repeatedly throughout the paper?

3. Data asking is explained well, but for what kind of information masking is needed? Take some example scenarios and include the same is paper, may be it will interest readers.

4. Why are you using fancy notations of Table 1 in Algorithm1 and 2.

Overall, it is very difficult to follow the paper and there is no flow of presentation. There is nothing related to tools used for implementing the proposed method. Lot many aspects are missing.

Author Response

Thank you for your valuable feedback, we have changed the manuscript just as suggested. We have tried to reply and explain all the valuable feedback point by point in the attached PDF. 

Reviewer 2 Report

In this paper, Physically Unclonable Functions (PUF) has been used to provide privacy of the healthcare device by masking the data and a machine learning technique is used for retrieving it back and to reduce security breaches on networks. The work is interesting, and there are some suggestions for further improvement.

Main comments:

1. In the experiment section, only the temperature data is used to verify the system’s accuracy. However, in practice, there are many kinds of health indicators, including heart rate [1], pulse, motion data [2], etc. The feasibility of various data masking and retrieving is not verified in this paper.

[1] Lbrini S, Fadil A, Aamir Z, et al. Big Health Data: Cardiovascular Disease Prevention Using Big Data and Machine Learning[M]//Machine Intelligence and Data Analytics for Sustainable Future Smart Cities. Springer, Cham, 2021: 311-327.

[2] Zhao H, Xu H, Wang Z, et al. Analysis and evaluation of hemiplegic gait based on wearable sensor network[J]. Information Fusion, 2023, 90: 382-391.

2. It is not clear whether the proposed model needs to establish one certain model for each patient. For example, the temperature of different people may vary over time. This paper does not address the applicability of the technology for the dataset corrected from different patients.

3. Please specify the Machine Learning model used, deep learning-based, neural network-based, or just machine learning-based model?

4. The paper can be better organized to avoid repeating some concepts and to avoid including unnecessary details. Examples are as follows:

(1) The first three sections can be reorganized into 1-2 sections.

Section 1. Introduction

Section 2. IoMT and Smart E-Healthcare

Section 3. Related Research

(2) At the beginning of Section 4, subsections 5.01-5.04 can be organized into one section, e.g., Section 5.1.

(3) Page 12, lines 371-377: When describing the 1st and 2nd models, please focus on the difference and improvement.

Minor points:

1. Define abbreviations and acronyms the first time they are used in the text, even after they have already been defined in the abstract.

(1) Using acronyms/abbreviations (e.g., PUF) in the paper title is not suggested.

(2) IoT in line 21 is not defined.

(3) AI/ML in line 83 is not defined.

(4) ICT in line 84 is not defined.

2. Avoid Long Sentences, as the reader may be bogged down with long, complex sentences, such as

(1) Technology mainly based on the IoT eliminates human mistakes and assists physicians in diagnosing diseases more quickly and precisely by integrating all essential parameter monitoring equipment through a connection with a systematic approach [2].

(2) The importance of individual and device identification to a network is that it guarantees that data is accurately ascribed and that data in the networks is only available to authorized personnel.

(3) Any IoMT smart devices are collection of different sensors and electronic circuits which helps to get the biomedical singles from the patient, network connectivity to send biomedical

data via a network, and a base-band processor to analyze biomedical signals a short or long term storage unit, a display platform with artificial intelligence systems for making decisions based on the doctor’s availability [8].

3. There are minor language mistakes and typos, such as

(1) Line 7: Physically Unclonable Functions (PUF) has been used toPhysically Unclonable Functions (PUFs) have been used to

(2) Line 7: In this “proposed” paper

(3) Line 56: which helps to get the biomedical “singles” from the patient

(4) Line 217: the closeness “among” two replies

(5) Line 217: “Intradistance” is the closeness among two replies to the same challenge…

(6) Line 55: Any IoMT smart “devices are collection” of

4. When writing a non-restrictive clause, place a comma before which, such as

(1) Line 10: this technique has exhibited 99.72% accuracy, which proves that this technique could actually be used to secure the healthcare data with masking.

(2) Line 55: Any IoMT smart devices are collection of different sensors and electronic circuits, which helps to …

Author Response

(The authors gave the same response as above.)

Reviewer 3 Report

The article has high quality. Achieved results should be printed. 

Author Response

Thank you for your valuable feedback, we have changed the manuscript just as suggested. 

Round 2

Reviewer 1 Report

I am not convinced by the explanation by the authors. The article does not provide enough technical justification related to machine learning model.

Author Response

Reviewers comment - I am not convinced by the explanation by the authors. The article does not provide enough technical justification related to machine learning model.

Authors reply - Thank you so much for your valuable feedback. We have added technical explanation related to machine learning in the manuscript. Thank you again for spending time to guide to improve the manuscript.

Reviewer 2 Report

The paper has improved considerably. I am happy that most of my comments have been professionally took into account.   The reviewer has no further questions.

Author Response

Reviewers comment - The paper has improved considerably. I am happy that most of my comments have been professionally took into account.   The reviewer has no further questions.

Authors comment - Thank you so much for your valuable feedback and thank you for spending you time to make this manuscript better.